# 3D Object Manipulation in a Single Image using Generative Models

## Abstract

Object manipulation in images aims to not only edit the object presentation but also gift objects with motion. Previous methods encountered challenges in concurrently handling static editing and dynamic motion applications, while also struggling to achieve realism in object appearance and scene lighting. In this work, we introduce **OMG3D**, a novel framework that integrates the precise geometric control with the generative power of diffusion models, thus achieving significant enhancements in visual performance. Our framework converts 2D objects into 3D, enabling user-directed modifications and lifelike motions at the geometric level. To address texture realism, we propose CustomRefiner, a texture refinement module that pretrain a customized diffusion model to align the style and perspectives of coarse renderings with the original image. Additionally, we introduce IllumiCombiner, an lighting processing module that estimates and adjusts background lighting to match human visual perception, resulting in more realistic illumination. Extensive experiments demonstrate the outstanding visual performance of our approach in both static and dynamic scenarios. Remarkably, all these steps can be done using one NVIDIA 3090. The code and project page will be released upon acceptance of the paper.

## 1 Introduction

Object manipulation seeks to modify or animate specific objects in an image to achieve enhanced visual effects. It has been widely utilized across various fields, such as poster design, AR/VR, and the film industry. Some advanced tools like PhotoShop offer pixel-level editing capabilities, including object addition and removal, lighting adjustments, and animation creation. However, their complexity often poses a significant barrier for beginners. As a result, many generative methods have emerged to offer more accessible and user-friendly alternatives to traditional editing tools.

Through analysis of existing methods, we identify two key limitations in object manipulation: (i) *Difficulty in concurrently handling static editing and dynamic motion application.* Current static editing methods, such as text based methods (Tumanyan et al., 2023; Hertz et al., 2022), effectively modify object appearances, and 3D reconstruction based methods (Chen et al., 2024b; Yenphraphai et al., 2024) offer precise control. But when applying these methods to animation, frame-by-frame editing fails to maintain consistency in both appearance and movement of objects, resulting in abrupt shifts. (ii) *Lack of realism in object appearance and scene lighting.* During static editing and motion application, generative methods often cause distortions in the object's presentations, yielding unpredictable outcomes. And methods use 3D reconstruction (Liu et al., 2023; Wu et al., 2024) falls short in delivering an accurate texture representation relative to the original image. Additionally, these methods typically overlook real lighting effects, lacking precise control over this aspect, resulting in a lack of visual realism in generated images and videos.

Considering the aforementioned limitations, we propose corresponding solutions to comprehensively achieve object manipultion. **To tackle the inconsistency between static editing and dynamic motion application**, we find that both tasks can be accomplished by binding skeletons to 3D geometric data and deforming them in 3D space. Thus, our framework, **OMG3D**, allows users to interact with reconstructed models, thereby achieving predictable outcomes in object manipulation. And by utilizing a graphics rendering pipeline, our framework ensures consistency in object appearance and smoothness in animation during the rendering process. **For addressing the lack of**

Figure 1: Overview of our proposed method. The teaser image provides a visual summary of the key contribution of this work.

**realism in texture and lighting**, we optimize textures of rough 3D models through differentiable rasterization (Laine et al., 2020). Additionally, we estimate (Zhan et al., 2021; Phongthawee et al., 2024) and process background lighting, delivering in a panoramic spherical light output. By integrating the optimized models with accurately corrected lighting into the rendering process, our framework can achieve lifelike and visually impressive outcomes.

To further demonstrate the efficiency of OMG3D, we explore specific modules designed to refine texture quality and enhance illumination accuracy. We first introduce **CustomRefiner**, which develops a customized diffusion model for each concept. After DDIM Inversion (Song et al., 2020) converts the image into Gaussian noise, the customized model (Ruiz et al., 2023) helps preserve the object's color and appearance while generating multiple viewpoints. During the denoising process, we incorporate depth control (Mahapatra & Kulkarni, 2022) and feature injections (Tumanyan et al., 2023) to maintain geometric details. Finally we fine-tune the texture using differentiable rasterization across multiple refined rendering viewpoints. And for realistic illumination, we introduce **IllumiCombiner**, which estimates background lighting and derives a spherical light source. To address the object's reduced color contrast rendered with estimated lighting, we develop strategies for light color correction and intensity enhancement, ensuring that the estimated lighting accurately reflects the object's original colors and saturation while maintaining the desired shadow direction.

Our proposed framework, OMG3D, is capable of performing a wide range of object manipulation tasks, including editing, composition, and animation. It outperforms methods in both 2D image editing and image-to-video generation. Recognizing the limitations of automatic quantitative metrics, we also conduct a user evaluation study, which shows that OMG3D's outputs are preferred over baseline results across cases. Compared to previous methods, our approach provides more physically accurate control over manipulations, along with higher-quality reconstruction results more realistic dynamic effects and lighting. In summary, our contributions are:

- The key contribution of OMG3D lies in proposing a unified and extensible framework, where advanced techniques are adapted, redesigned, and integrated to address the unique challenges both static and dynamic generation with high fidelity. Through 3D object manipulation, OMG3D bridging the gap between static transformations and temporal dynamics.

- We introduce CustomRefiner, a texture optimization module that utilizes a custom diffusion model to adjust the style and viewpoints of coarse renderings, ensuring texture alignment with the original image.

- We introduce IllumiCombiner, which estimates and processes background lighting in images, achieving realistic light and shadow effects.

- We conduct extensive experiments to demonstrate the superiority of our method compared to existing methods.

## 2 RELATED WORK

**Image editing.** Previous methods like GANs (Goodfellow et al., 2020; Karras et al., 2019; 2020) have broadened the concept of image editing into various topics, including text-based manipulation

(Shen et al., 2020; Wu et al., 2021) , image-to-image translation (Isola et al., 2017; Zhu et al., 2017), latent manipulation (Shen et al., 2020; Wu et al., 2021), and style transfer (Gatys, 2015).

The rise of diffusion models (Ho et al., 2020; Peebles & Xie, 2023; Rombach et al., 2021; Podell et al., 2023) has sparked a new wave of interest in image editing, exploring ways to enhance pre-trained diffusion models with text-guided editing capabilities (Brooks et al., 2023; Hertz et al., 2022; Mokady et al., 2023; Tumanyan et al., 2023; Zhou et al., 2024b;a). And Attention modification methods enhance the operations of attention layers (Vaswani, 2017), providing a common and straightforward approach to training-free image editing. Both MasaCtrl (Cao et al., 2023) and PnP (Tumanyan et al., 2023) prioritize the replacement of attention features to improve consistency. Specifically, MasaCtrl targets the Key and Value components in the self-attention layer, whereas PnP concentrates on the Query and Key elements within the same layer.

To achieve more accurate control, ControlNet (Zhang et al., 2023a) incorporates additional conditional inputs such as depth (Yang et al., 2024a), poses (Cao et al., 2017), and edges (Xie & Tu, 2015) for image generation. Similarly, some methods utilize geometry data for image editing and other tasks (Zhao et al., 2024). OM3D (Kholgade et al., 2014) matches objects in images with existing 3D assets, enabling editing through control of these 3D assets. Recently, methods like OBJect3DIT (Michel et al., 2024) have achieved 3D-aware editing using language instructions, but their capabilities are limited due to training on synthetic data. Image sculpting (Yenphraphai et al., 2024), a baseline of our paper, struggles to produce realistic lighting and shadow effects.

**Image Animation.** Animating static images has been studied for a long time. Previous methods focus on simulating motion for natural dynamics (Holynski et al., 2021; Jhou & Cheng, 2015; Mahapatra & Kulkarni, 2022), as well as human faces (Geng et al., 2018; Wang et al., 2022a) and bodies (Weng et al., 2019; Siarohin et al., 2021). And video prediction methods (Franceschi et al., 2020; Zhang et al., 2020) generate future frames from single images by extracting spatial-temporal patterns from video data. Additionally, some techniques (Endo et al., 2019; Zhao & Zhang, 2022; Voleti et al., 2022) integrate motion priors into their pipelines to achieve more consistent video outputs.

With the rise of Video Diffusion Models (Ho et al., 2022), numerous methods (Wang et al., 2024a; Luo et al., 2023; Wang et al., 2023) utilize images as conditions to generate videos. Previous methods (Shi et al., 2024; Mahapatra et al., 2023) utilize estimated optical flow in conjunction with a pre-trained text-to-image diffusion model (Rombach et al., 2021) to achieve video generation. Some concurrent works focus on controllable editing through motion conditions, including object movement (Chen et al., 2024a), camera movement (Yang et al., 2024b), and methods for editing (Bai et al., 2024). However, these approaches do not support region-specific image-to-video (I2V) transformations and rely on standard 1-D temporal attention for learning the complex mapping from images to videos. Due to the limitations of model capabilities, image-to-video techniques still have significant room for improvement in maintaining both object appearance and motion fluidity.

**Image harmonization.** Image harmonization is a crucial vision task that ensures objects are naturally integrated into new backgrounds, and many studies have focused on addressing this challenge. (Wang et al., 2022b) proposed a method that first estimates an HDR sky map and then employs Differentiable Object Insertion to blend virtual objects with the background, though their work is primarily focused on outdoor scenes. (Careaga et al., 2023) adjusted the foreground albedo to align with the background and estimated environmental lighting to refine the shading process. However, obtaining accurate albedo information for objects can be challenging, and their method struggles to model the cast shadows that objects may produce in the new environment. (Enyo & Nishino, 2024) introduced a technique that utilizes 3D object information to generate high-quality Reflectance Maps, enabling the recovery of detailed lighting information. However, for objects without specular reflection, this model may be less effective in recovering high-frequency lighting details. Methods such as DiffusionLight (Phongthawee et al., 2024) and DiPIR (Liang et al., 2024) effectively estimate environmental lighting from background images.

Our framework is designed to enable seamless combinations of arbitrary foregrounds and backgrounds while maintaining consistency in shadows and lighting. DiPIR, which requires per-object optimization for environmental lighting and is less suitable for our application, our approach emphasizes plug-and-play flexibility. Additionally, methods like DiffusionLight and DiPIR could serve as alternative implementations for the lighting estimation module in our framework, further enhancing its versatility and robustness.

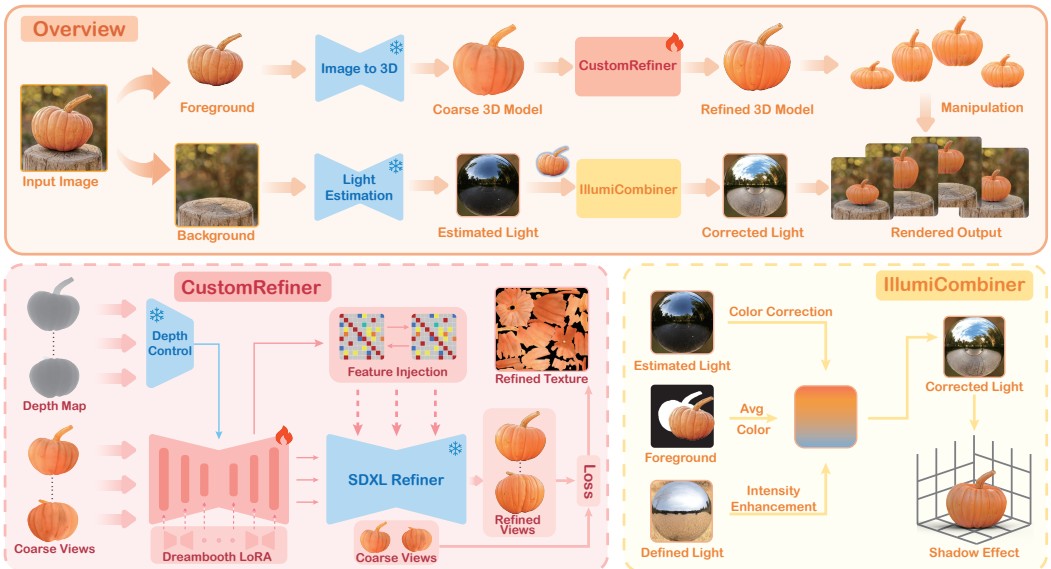

Figure 2: We demonstrates the workflow of OMG3D (§4.1). OMG3D has two key modules: (1) CustomRefiner (§4.2), and (2) IllumiCombiner (§4.3). The former leverages a customized diffusion model to optimize the rendering viewpoints and subsequently improve the model's textures. The latter uses color information from the original image and predefined light sources to correct the color and enhance the intensity of the estimated light, ensuring natural lighting in the final image.

## 3 PRELIMINARY

**Diffusion models.** Diffusion probabalistic models (DPM) (Ho et al., 2020; Song et al., 2020; Zhao et al., 2023) aims to predict a data distribution $p_{\text{data}}$ through a progressive denoising process. It is represented as a discrete-time stochastic process $\{x_t\}_{t=0}^{T}$ where $x_0 \sim p_{\text{data}}$, and $\mathbf{x}_t \sim \mathcal{N}(x_{t-1}; \sqrt{\alpha_t/\alpha_{t-1}}x_{t-1}, (1 - \alpha_t/\alpha_{t-1})I)$. The decreasing scalar function $\alpha_t$, with constraints that $\alpha_0 = 1$ and $\alpha_T \approx 0$, controls the noise level through time. It can be shown that

$$x_t = \sqrt{\alpha_t}x_0 + \sqrt{1 - \alpha_t}\epsilon, \text{ where } \epsilon \sim \mathcal{N}(\mathbf{0}, I). \tag{1}$$

A diffusion model $\epsilon_\phi$ is trained to predict the noise $\epsilon$ from $x_t$, the simplified training loss is

$$\mathcal{L} = \mathbb{E}_{x_0, t, \epsilon}\|\epsilon_\phi(\sqrt{\alpha_t}x_0 + \sqrt{1 - \alpha_t}\epsilon, t, C) - \epsilon\|_2^2, \tag{2}$$

where $C$ denotes conditioning signals such as text or images.

**Image to 3D.** DreamFusion (Poole et al., 2022) generates 3D models from text by leveraging a pretrained diffusion model $\epsilon_\phi$ as an image prior, optimizing the 3D representation parameterized by $\theta$. Building on this approach, in image-to-3D generation, the diffusion model $\phi$ is trained to predict the sampled noise $\epsilon_\phi(x_t; x_r, t)$ of the noisy image $x_t$ at the noise level $t$, conditioned on the reference image $x_r$. A *score distillation sampling* (SDS) loss encourages the rendered images to align with the distribution modeled by the diffusion model, where $\omega(t)$ is the weighting function:

$$\nabla_\theta \mathcal{L}_{\text{SDS}}(\phi, g(\theta)) = \mathbb{E}_{t, \epsilon}\left[\omega(t)(\epsilon_\phi(x_t; x_r, t) - \epsilon)\frac{\partial x}{\partial \theta}\right], \tag{3}$$

## 4 METHOD

We present a novel diffusion model based framework, OMG3D, for comprehensive object manipulation from images (see Fig. 2). In the following sections, we will provide an overview of the framework in Sec. 4.1, introduce the texture optimization module CustomRefiner in Sec. 4.2 and the lighting processing module IllumiCombiner in Sec. 4.3.

### 4.1 OVERVIEW OF OMG3D

Given a 2D image, our framework is designed to enable manipulation of the objects within a 3D space, encompassing both presentation edits and motion applications, before eventually rendering them back into a 2D image. This process unfolds in three key steps: 1. Converting the 2D object into a 3D model and optimizing its texture, 2. Deforming the reconstructed model within 3D space, 3. Rendering the modified model with enhanced lighting to improve the visual output.

**3D model reconstruction.** Given an image of an object, we first isolate the object using methods like SAM (Kirillov et al., 2023). We then use the object segmentation as a reference to reconstruct the mesh through Image-to-3D methods (Liu et al., 2023; Wu et al., 2024; Mildenhall et al., 2021; HyperHuman, 2024; Zhang et al., 2023b; 2024; Xu et al., 2023; Long et al., 2023). Currently, the reconstructed results often display a considerable gap in texture presentation compared to the original image. To address this, we apply **CustomRefiner** to enhance the texture, significantly improving the fidelity of the final output.

**3D model deformation.** After optimizing the 3D model, users can manually construct a skeleton and interactively manipulate it by rotating the bones to achieve the desired pose or attach animations to the skeleton, using 3D manipulation software such as Blender. Because the texture mapping of a 3D model is exclusively linked to the vertices and independent of the pose, the appearance of the model remains consistent throughout this process.

**Rendering results.** Our framework utilizes a graphics rendering pipeline, which inherently provides the advantage of utilizing light sources effectively. Consequently, we focus on estimating and processing the lighting from the background image to derive an accurate spherical light thorough **IlluminCombiner**. When we obtain the deformed model and processed light, we render the results within the graphics rendering pipeline to achieve realistic and visually balanced effects.

### 4.2 CROSS-VIEWPOINTS GENERATIVE TEXTURE REFINEMENT

To address the gap between rough models reconstructed from image-to-3D methods and the original image, we propose **CustomRefiner**, a texture refinement module that utilizes differentiable rasterization (Laine et al., 2020) across multiple refined rendering viewpoints. Specifically, our module focuses on three key aspects: 1.Appearance and color preservation, 2.Geometry information injection, and 3.UV-texture optimization.

**Appearance and color preservation.** Our method draws inspiration from customized models, retaining the original object's appearance and color when generating new viewpoints. Initially, we train a customized diffusion model like DreamBooth (Ruiz et al., 2023) for each concept. And we convert various rough rendering viewpoints back to Gaussian noise using DDIM Inversion (Song et al., 2020). During the denoising stage, we leverage information preserved by the pretrained customized model to guide these viewpoints into visual alignment with the original image.

**Geometry information refinement.** Since we can easily obtain precise depth maps of rough models through the graphics rendering pipeline, we incorporate Depth ControlNet (Mahapatra & Kulkarni, 2022) into the DDIM Inversion stage and diffusion denoising stage to ensure consistency in shape and geometric information. We directly add the depth information to these two stages without any fine-tuning. Additionally, to further enhance the preservation of geometric details, we draw inspiration from PnP (Tumanyan et al., 2023). During the inversion stage, we save the feature maps from the residual blocks and the self-attention maps from the transformer blocks. Then, in the denoising stage, we replace these features with the saved maps, ensuring that more geometric information is preserved during the refinement process.

**UV-texture optimization.** Through DDIM Inversion (inverting coarse rendering viewpoints into gaussian noise), incorporating Depth ControlNet with feature injection, we achieve optimized results across various viewpoints. We achieve gradient backpropagation directly to the UV texture map through differentiable rasterization, optimizing with the following pixel-wise MSE Loss:

$$\mathcal{L}_{\text{MSE}} = ||I_{\text{fine}}^p - I_{\text{coarse}}^p||_2^2 \tag{4}$$

where $I_{\text{fine}}^p$ and $I_{\text{coarse}}^p$ are the object rendering viewpoints before and after optimization at the predefined camera pose $p$.

Table 1: Quantitative comparison on image editing against other methods (§5.1).

| Method | GPT-4o | | User Study | |
|---|---|---|---|---|
| | Image Align↑ | Text Align ↑ | Image Align↑ | Text Align ↑ |
| P2p (Hertz et al., 2022) | 2.00 | 1.00 | 1.84 | 1.92 |
| Masactrl (Cao et al., 2023) | 3.20 | 1.80 | 2.7 | 2.03 |
| Pnp (Tumanyan et al., 2023) | **4.80** | 2.00 | 3.68 | 2.18 |
| ImgScu(Yenphraphai et al., 2024) | 4.00 | 3.00 | 3.77 | 3.34 |
| **Ours** | **4.80** | **4.80** | 3.93 | **4.38** |

### 4.3 HYBRID LIGHTING PROCESSING

Achieving realistic lighting and shadow effects has long been a challenge that many methods seek to address. Yet our framework enhances these effects by incorporating spherical light sources to achieve superior results. To further enhance this capability, we propose IllumiCombiner, a lighting processing module that delivers realistic and visually balanced spherical light. Specifically, the process involves two steps: 1. estimating and processing the lighting from background images, and 2. creating a transparent plane for accurate shadow reception.

**Light processing.** We explore various lighting estimation methods (Zhan et al., 2021; Phongthawee et al., 2024) to assess the background lighting. These methods export estimation as a spherical light $L_e$, which has two properties: **color component** $c_e$ **and intensity component** $i_e$. However, we find that images directly rendered with $L_e$ often exhibit reduced color, even with accurate lighting direction. For color correction, we use SAM (Kirillov et al., 2023) to segment the image $i$, obtaining a mask $M$. We then calculate the average color within $M$, resulting in object's color $c_a$.

$$c_a = \frac{1}{|M|} \sum_{x \in M} i(x) \tag{5}$$

Where $|M|$ represents the number of pixels in the mask $M$, and $i(x)$ represents the color value of pixel $x$ in the image $i$. We design a color-blending coefficient $\lambda_1$ and blend the estimated lighting color $c_e$ with $c_a$ with to correct the color of the estimated lighting. Here $c_{ec}$ is the corrected color.

$$c_{ec} = \lambda_1 \boldsymbol{c}_e + (1 - \lambda_1)c_a \tag{6}$$

For intensity enhancement, we aim to maintain a base level of light's brightness, preventing the rendered image from appearing too dark due to insufficient estimated intensity and ensuring satisfactory color saturation. To achieve this, we define a uniform ambient light $L_d$ (with equal intensity $i_d$ in all directions and uniformly white in color), which inherently provides a consistent level of brightness. We then design a blending coefficient $\lambda_2$ to mix the $i_e$ and $i_d$, ensuring object's saturation through the defined light while retaining the shadow direction provided by the estimated light. The final enhanced intensity $i_{ec}$ is:

$$i_{ec} = \lambda_2 i_e + (1 - \lambda_2)i_d \tag{7}$$

**Plane creation.** In the rendering process, a physical entity is required to capture shadows. To enhance the natural appearance of the shadows cast by the light source, we utilize Depth Anything to estimate the ground depth in the background image. This allows us to approximate the plane's normal vector from the depth map. With this normal vector, we can automatically create a transparent plane in Blender to capture the shadows without rendering the plane itself. By refining the rough model and applying precise manipulation over lighting and shadow effects, our approach enables us to render more realistic and visually balanced results.

## 5 EXPERIMENTS

**Experimental Setup.** We follow previous image-to-3D methods (Liu et al., 2023; Wu et al., 2024; HyperHuman, 2024) to get texture 3D model. During the texture refinement process, we use Dreambooth (Ruiz et al., 2023) with SDXL-1.0 (Podell et al., 2023) to implement a customized diffusion model. Specifically, we fine-tune the customized model using LoRA (Hu et al., 2021) for 3000 steps with a learning rate of 3e-5. During the denoising process, we incorporate the Depth ControlNet for depth control (Mahapatra & Kulkarni, 2022). In the feature injection stage, we leverage all the self-attention layers of the SDXL decoder and the first block of the SDXL's upsampling decoder.

Table 2: Quantitative comparison on image animation against other methods (§5.1).

| Method | GPT-4o | | User Study | | | |
|---|---|---|---|---|---|---|
| | Image Align ↑ | Text Align ↑ | Image Align ↑ | Text Align ↑ | Realism ↑ | Consistency ↑ |
| SVD (Blattmann et al., 2021) | 4.0 | 2.0 | 3.68 | 2.25 | 2.53 | 2.52 |
| Pika (pika, 2024) | **5.0** | 4.0 | **4.07** | 1.78 | 3.48 | 3.37 |
| DynamiCrafter (Xing et al., 2023) | **5.0** | 4.0 | 3.75 | 1.68 | 2.87 | 2.85 |
| ImgScu(Yenphraphai et al., 2024) | 3.0 | 4.0 | 2.88 | 2.4 | 1.85 | 2.02 |
| **Ours** | **5.0** | **5.0** | 4.03 | **4.65** | **4.03** | **4.4** |

For object deformation and animation, we utilized some automated methods, such as Mixamo (mixamo, 2024) and SINMDM (Raab et al., 2023), to complete the rigging of the model's movements and enhance the range of actions. For example, we use platforms like (mixamo, 2024) to upload meshes online, select skeletal key points, and let the system automatically rig the skeleton and generate corresponding animations based on the selected actions. Moreover, users can also create their own animations. Additionally, we employ DiffusionLight (Phongthawee et al., 2024) as the foundational model in our IllumiCombiner. During the color correction and intensity enhancement phases, we set $\lambda_1$=0.5 and $\lambda_2$=0.5. For background inpainting, we use Adobe Firefly (Adobe, 2023) to ensure high-quality results. Notably, all experiments are completed on a single NVIDIA RTX 3090 GPU with 24GB vRAM.

**Metric.** Existing evaluation methods like FID (Heusel et al., 2017) require large dataset to be effective in image editing tasks, and the metrics often do not correspond well with the human perceptions. To better assess the effectiveness of our method, we utilize the current state-of-the-art vision-language model, GPT-4o (GPT4, 2024), to evaluate our edited results. The evaluation focuses on the following aspects: whether the result aligns with the text description and whether the edited object's appearance remains consistent with the original. Similarly, for video evaluation, there is a lack of effective metrics for assessing image animation. Therefore, we sample video frames and use GPT-4o to evaluate the consistency of the object's appearance throughout the animation. Recognizing the limitations of automatic quantitative metrics, we also conduct a user study to assess the actual visual quality of our method, including the realism and consistency of the video motion.

## 5.1 EVALUATION

**Quantitative Results.** To demonstrate the effectiveness of our method, we compare it against several baselines that are capable of achieving object manipulation. For the image editing task, we select P2P (Hertz et al., 2022), PnP (Tumanyan et al., 2023), and Masactrl (Cao et al., 2023) for comparison. While for the image animation task, we choose SVD (Blattmann et al., 2023), Pika (pika, 2024), and DynamiCrafter (Xing et al., 2023) as video baselines. In our evaluations, we found that GPT4o to some extent demonstrates the capabilities of editing methods, but it tends to have higher tolerance when it comes to object appearance alignment.

As shown in Tab. 1, in the image editing task, our method achieves excellent results in both GPT-4o and human evaluations. It performs the best in terms of aligning with the original appearance of the object as well as adhering to the textual descriptions. In the video generation task, our method not only ensures a high degree of consistency in maintaining the object's appearance but also successfully generates the target actions, shown in Tab. 2.

**Qualitative Results.** We present the image editing comparison results in Fig. 3. Traditional text-based editing methods struggle to ensure the consistency of the object's appearance and also have difficulty executing complex editing commands, such as changes in the object's pose, quantity, and position. While Image Sculpting can achieve these goals to some extent, because it optimizes the rendered images, **it often results in edge blurring and details loss (e.g., the missing elephant leg in the second row, fifth column, and the missing basketball for the right toy in the fourth row, fifth column).** Additionally, these methods typically lack realistic shadow effects, which compromises the final visual quality and realism.

For video editing, although I2V generation methods can maintain the appearance of the object, they struggle to generate the desired actions. As shown in Fig. 5, in the videos generated by SVD and Pika, the object remains stationary throughout, and the object's appearance becomes distorted (highlighted in the first and second rows). While the latest method, DynamiCrafter, can produce some

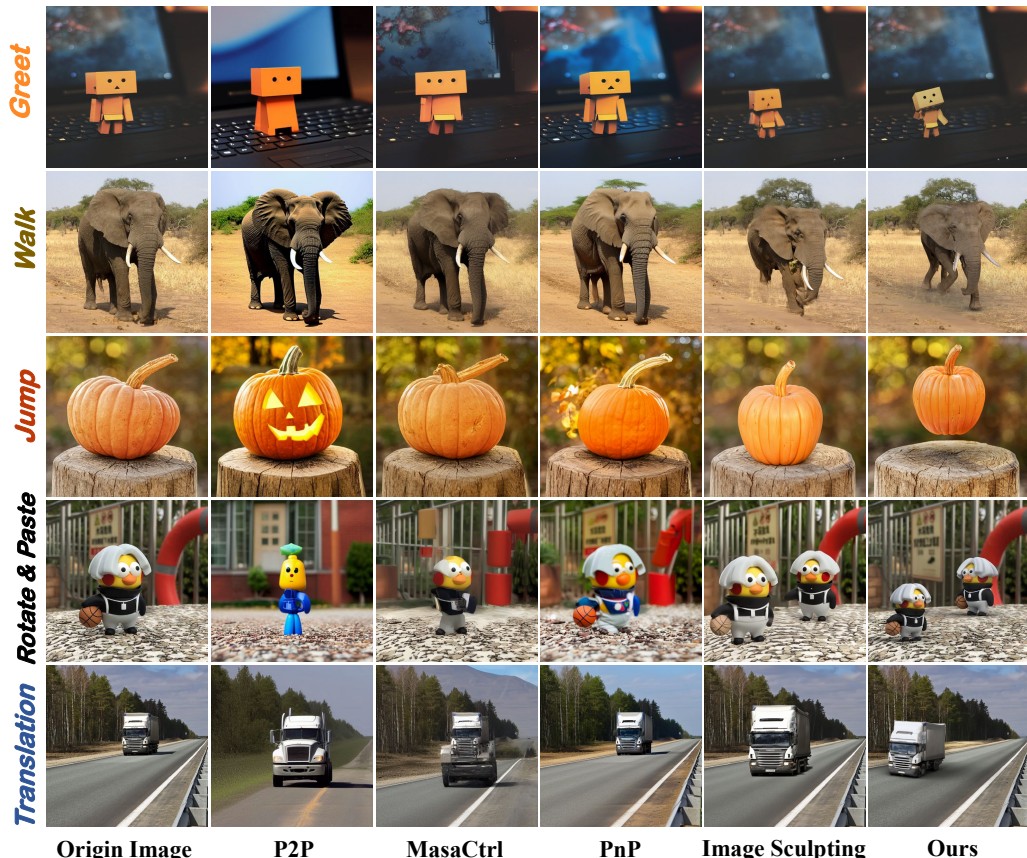

Figure 3: Qualitative comparison with image editing methods (§5.1). Text-based methods fail to achieve the target action and maintain appearance, while Image Sculpting faces partial object loss.

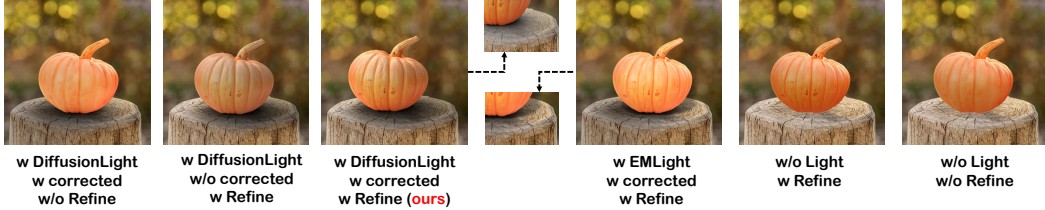

Figure 4: Ablation study (§5.2) on whether to use CustomRefiner and IllumiCombiner, different light estimation methods, and the presence or absence of shadows. W corrected means we use IllumiCombiner, and w Refine means we use CustomRefiner.

motion, it fails to align with the text prompt and introduces artifacts (highlighted in the third row). When using the Image Sculpting method to edit each frame individually to generate a video, it fails to ensure the consistency of the object's appearance, the smoothness of the actions and **totally no shadows**, resulting in incorrect outputs (highlighted in the fourth row). In contrast, our method not only ensures the consistency of the object's appearance but also generates smooth video sequences that align with the text descriptions, achieving a superior visual outcome.

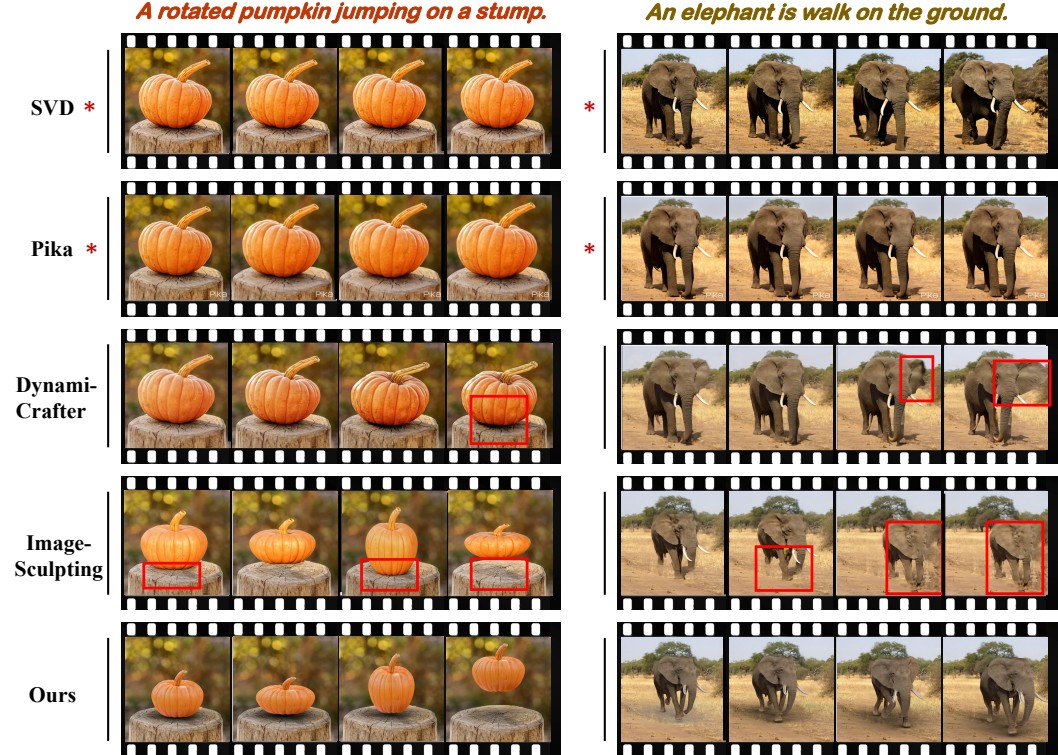

Figure 5: Qualitative comparison with image animation methods (§5.1). * indicates failure cases where the object remains stationary in the video. Distorions are highlighted by red rectangles. The pumpkin's positions in Image Sculpting are changed across frames.

## 5.2 ABLATION STUDY

**Impact of CustomRefiner**. We design a series of ablation studies to evaluate the effect of using CustomRefiner in OMG3D. As shown in Fig. 4, the use of CustomRefiner not only enhances the object's color but also increases image details, improving the realism of the object's appearance.

**Impact of light estimation methods and IllumiCombiner.** We compared two different light estimation methods, EMLight (Zhan et al., 2021) and DiffusionLight (Phongthawee et al., 2024). From Fig. 4, we can see that the results rendered with EMLight exhibit overexposure and incorrect shadow direction. In close-up comparisons, DiffusionLight produces more realistic shadows, while EMLight's shadows have harsh and inconsistent edges. By integrating color information from the original image and adjusting the intensity and hue of the estimated light, IllumiCombiner ensures that the lighting in the scene aligns more naturally with the background, leading to a more harmonious and aesthetically pleasing final output.

**Impact of shadow inclusion.** As illustrated in Fig. 4, without shadows, the object appears flat and lacks realism, making it seem disconnected from the surrounding environment. When shadows are added, the object not only blends more naturally with the background but also gains a sense of depth and dimensionality. The inclusion of accurate shadows helps anchor the object within the scene, enhancing the overall visual coherence and realism.

## 5.3 APPLICATIONS

Our method is not only capable of performing image editing and image animation but also allows the refined model and corrected lighting to be seamlessly combined with any background image, as shown in Fig. 6. This flexibility makes our approach particularly valuable in fields such as virtual reality (VR) and augmented reality (AR). In these applications, maintaining seamless integration

between objects and environments, as well as realistic lighting, is crucial. Our technique ensures not only that the object's appearance is consistent with the background but also that it supports dynamic interaction, thereby enhancing user immersion and visual experience.

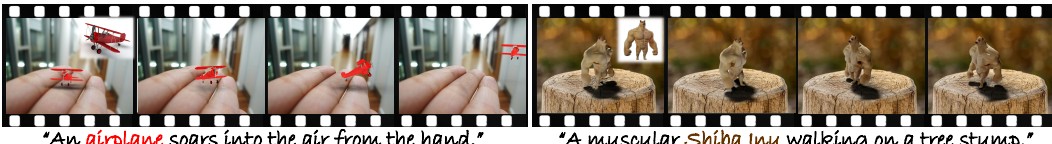

"An *airplane* soars into the air from the hand."     "A muscular *Shiba Inu* walking on a tree stump."

Figure 6: Applications of OMG3D (§5.3): combine different models with background images.

## 6 CONCULUTION

In this work, we presented **OMG3D**, a novel framework for object manipulation in images that not only enhances object presentation but also brings objects to life with dynamic motion. OMG3D seamlessly integrates precise geometric control with the powerful capabilities of diffusion models, offering significant improvements in visual quality. By converting 2D objects into 3D models, our framework allows user-directed manipulations while preserving object appearance with high fidelity. The introduction of CustomRefiner enables detailed texture refinement, aligning the rendering with the original image's style and perspective. Meanwhile, IllumiCombiner provides advanced lighting adjustments, resulting in visually accurate and realistic illumination. Our experiments demonstrate that OMG3D excels in both static and dynamic scenarios, significantly outperforming previous methods in terms of visual realism, motion accuracy, and adaptability to various conditions. We show that our proposed framework outperforms state-of-the-art methods, enabling a wide range of applications. Future work will focus on extending our framework to more complex scenes and incorporating object dynamic interactions.

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

# A APPENDIX

## A.1 LIMITATION AND FUTURE WORKS.

Despite the success of our method in object manipulation, there exists some limitations. For example, the 3D-generated models show suboptimal mesh quality in complex cases, which could be improved with advanced models or manual adjustments.

About lighting: DiffusionLight sometimes produces inaccurate lighting, highlighting the need for better light estimation methods. We have analyzed some existing lighting techniques in our Sec. 2. However, in comparison to existing lighting methods, our IllumiCombiner provides a substantial improvement over more recent methods, such as DiffusionLight (CVPR 2024), as demonstrated in Fig. 4. We believe that our approach to lighting design strikes a balance between effectiveness and efficiency, tailored specifically to the objectives of our framework. Due to the plug-and-play design of our framework, we are also very excited about the possibility of new lighting methods emerging in the future, which could further improve the overall performance of our framework.

In future work, we aim to expand our method further. Currently, the background is static, but we plan to explore the use of video models to generate dynamic backgrounds or reconstruct 3D scenes for more complex environments. Additionally, we intend to incorporate dynamic interactions between objects, enabling the system to not only handle individual objects but also simulate complex interactions between multiple objects.

## A.2 MORE ABLATION STUDIES

We conduct ablation experiments on various 3D generation methods, including text-to-3D approaches using VSD loss like ProlificDreamer (Wang et al., 2024b), well-known open-source projects such as Zero123 (Liu et al., 2023) and Hunyuan (Yang et al., 2024c), as well as commercial models like HyperHuman (HyperHuman, 2024). The results demonstrate that our framework is versatile and can be effectively applied to a wide range of methods. While ProlificDreamer generates high-quality 3D results, it is designed for different tasks and is unable to precisely match the input image. We believe that as image-to-3D methods continue to evolve, our framework can seamlessly integrate these advancements to achieve even better results.

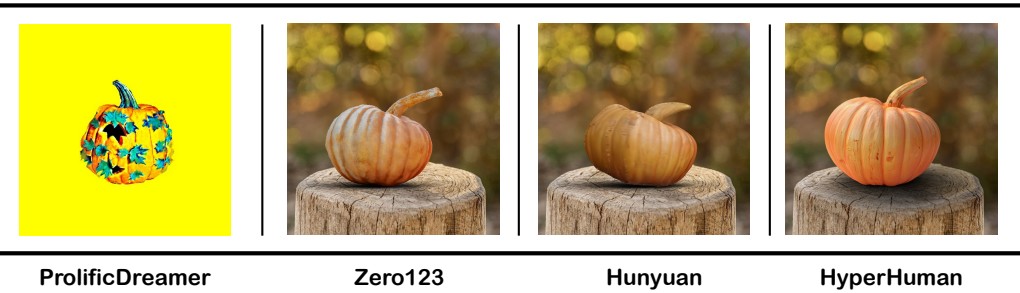

| ProlificDreamer | Zero123 | Hunyuan | HyperHuman |

Figure 7: Ablation study on different image to 3D generation methods.

We conduct ablation studies under different experimental techniques. To intuitively demonstrate the effects of these techniques, we select results generated from novel viewpoints for visualization. The results show that without using Dreambooth, the surface of the generated object appears rough. If depth control is not incorporated during the DDIM inversion or sampling process, the object's edges exhibit not smooth or interference from the background. Additionally, if injection is not used, it negatively impacts the quality of the generated image.

## A.3 DIFFERENCE WITH IMAGE SCULPTING.

Our motivation fundamentally differs from that of Image Sculpting(ImgScu). The core motivation of our work lies in two aspects: First, achieving realistic lighting and shadow effects in both static

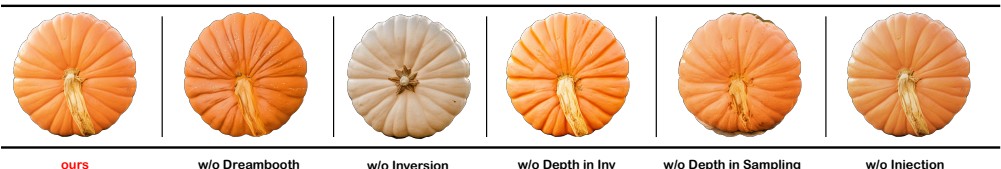

Figure 8: Ablation study on SDXL settings.

and dynamic generation; Second, ensuring consistent object appearance and smooth motion when applying 3D models directly in dynamic generation. In contrast, ImgScu primarily focuses on using 3D depth information to generate results and employing customized models to refine 2D outputs.

About methods, ImgScu relies solely on depth information during the generation process, which leads to issues such as edge blurring and loss of details. When applied to frame-by-frame editing, this inconsistency in object appearance can severely degrade video quality. Combined with the limitations in static editing, this results in visual artifacts like flickering. Our method directly renders the refined 3D object back to 2D. Once the texture optimization is complete, the refined 3D model can be directly rendered for both static and dynamic scenarios without requiring further refinement of the rendering results. The physical rendering process in our method ensures object appearance consistency and motion continuity, effectively avoiding issues such as blurring and detail loss. Moreover, the physical rendering process inherently supports the incorporation of light sources to achieve realistic lighting effects—a capability ImgScu lacks. This fundamental difference underscores the significant advantage of our method over ImgScu.

## A.4 MORE CASES

We provide additional demonstrations showcasing the application of our framework in human motion generation. These examples highlight the versatility and effectiveness of our approach in handling complex motion tasks while maintaining high fidelity and consistency.

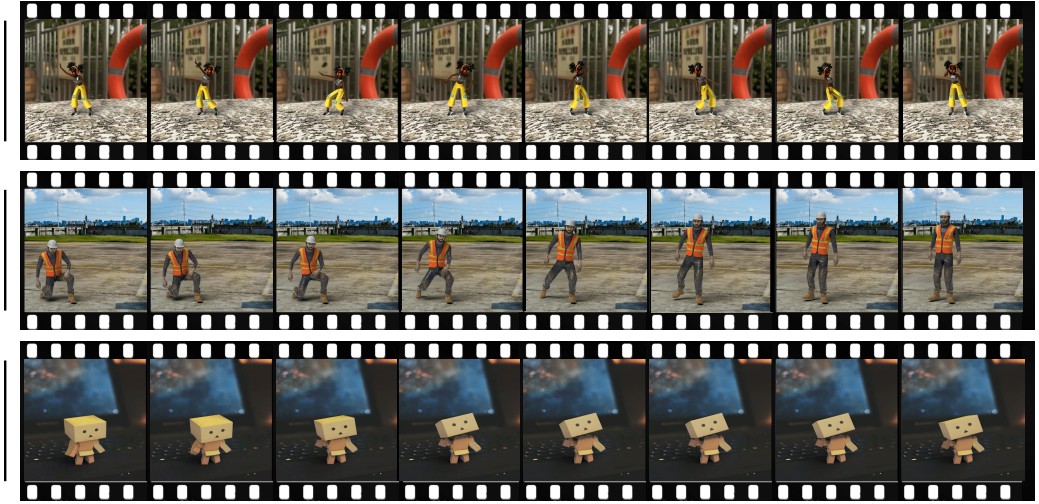

Figure 9: More applications.

| Origin Image | Incorrect mesh | Origin Image | Incorrect color | Incorrect light | Incorrect light |

Figure 10: Failure cases (§A.4).

We present several failure cases. First, the generated cat mesh deviates from the original, appearing slimmer than the actual physique. Second, the texture color differs significantly from the original, and even with the refinement module, correcting these discrepancies, especially with complex textures, is challenging. Third, the current light estimation methods struggle in some images, producing inaccurate lighting and shadows, leading to visual inconsistencies. More successful examples are shown in https://anonymous.4open.science/r/OMG3D-33D5/.

