# OpenReview forum: "3D Object Manipulation in a Single Image Using Generative Models"
_ICLR.cc/2025/Conference — Submitted to ICLR 2025_

### Official Review · Reviewer_LJb3 · 2024-11-02

**Soundness:** 3
**Presentation:** 4
**Contribution:** 2
**Rating:** 6
**Confidence:** 3

**Summary:**

The paper proposes OMG3D for 3D editing of objects. It takes an image and produce 3D assets with high quality texture and lighting estimation. Then 3D editing including motion and shadow effects can be added in common 3d editing softwares. To get high-quality results, it proposes a texture refinement and a lighting estimation module in the image-to-3d stage. The method is compared against image and video generation models using user study and LLM-based evaluations, under which it works better in realism and consistency axis, and works better or on-par in terms of image/text alignment.

**Strengths:**

- The paper is well written and the video demo is well made.
- The method produces high-quality results, and those can be plugged into existing graphics pipelines relatively easily due to the explicit modeling of geometry, texture and lighting.
- The authors did a good job in quantitative evaluations and comparisons, including some recent image/video generation methods.
- The method is lightweight and can run on a single 3090 GPU.

**Weaknesses:**

- Is the comparison to images/video models fair? For instance, the pipeline introduced in this paper involves human editing (e.g., rigging and animation) while the baselines does not. Although I like the fact that the comparisons reflect the final quality, the different level of human involvement can be made clearer and discussed.
- It would be useful to explicitly state the steps that require manual intervention and differentiate those from the automatic component of the method through a table. For instance, intensity enhancement needs manual adjustment but lighting estimation does not.
- Lack of ablations and comparison to image-to-3D methods. Since this is main technical contribution, it would be useful to ablate the design choices that are different from existing methods, and report quantitative results showing their individual contributions to the overall performance. For instance, how much does DDIM inversion help, how much does dreambooth customization help, how much does depth controlnet help, and how much does feature injection helps. Ideally, by ablating each of the designs one after another, it can reach a known baseline.

**Questions:**

- How is depth control incorporated? Is it added to SDXL during Dreambooth finetuning? What does the denoising function look like?

- How to deal with multiple planes (e.g., table and ground) in plane creation process? Since depth from Depth Anything is up to scale, how to adjust the plane to be in contact with the object?

- How is the material property specified during rendering in the editing stage?

---

> ### Author Response · Authors · 2024-11-20
> ****Rebuttal to Reviewer LJb3****
>
> We are grateful for your recognition of the technical soundness of our method and the effectiveness of our design choices. Your acknowledgment of our convincing results and well-structured presentation is highly appreciated.
>
> We will address your inquiries and concerns point by point in the following responses.
>
> **W1:** Is the comparison to images/video models fair?
>
> **Answer to weakness 1:**
> We have made every effort to ensure fairness. Most methods cannot incorporate 3D priors for control, whereas our motion sequences are generated automatically with minimal manual intervention.
> 1. In lines 301-303 of our old version PDF, we have described several methods for automating action binding. In the supplementary materials (line 776 of the old version PDF), we have provided a link to our anonymous Git repository, which demonstrates some human animation effects achieved through automated binding.
> 2. Our workflow consists of five main steps: 3D model generation, texture optimization, lighting processing, object manipulation, and final rendering. Among these, object manipulation can be accomplished **using existing automated motion binding methods or manually performed by professional modeling artists. Except for the predefined light sources, all lighting processes, including estimation and correction, are fully automated and implemented through code.**
>
> **W2:** Can you clarify which steps require manual intervention and which are fully automated, possibly through a table?
>
> **Answer to weakness 2:**
> 1. Thank you very much for your suggestion, visualizing the steps of our framework in both automated and manual formats would indeed enhance understanding. We have organized this information in the following table. Here action binding can be achieved through automated methods or manually completed by experienced professionals.
> | Steps   | 3D Generation | Texture Refinement | Action Binding | Light Processing  |
> |---------|----------------|--------------------|----------------|------------------|
> | Auto    | √              | √                  | √              | √                |
> | Manual  | -              | -                  | √              | √  (only pre-defined light needs)              |
>
> **W3:** Lack of ablations on image-to-3D methods and techniques used.
>
> **Answer to weakness 3:**
> Thank you for your valuable suggestions. **We add ablation studies about 3D generation methods** in Figure 7 of our new version PDF. And in Figure 8 (the supplementary material of the new version PDF), **we add ablation studies on the various techniques utilized in our framework** to evaluate the contributions of each technique. To intuitively demonstrate the effects of these techniques, we select results generated from novel viewpoints for visualization. The results show that without using Dreambooth, the surface of the generated object appears rough. If depth control is not incorporated during the DDIM inversion or sampling process, the object’s edges exhibit not smooth or interference from the background. Additionally, if DDIM inversion or injection is not used, it negatively impacts the quality of the generated image.
>
> **Q1:** How is depth control incorporated? Is it added to SDXL during Dreambooth finetuning? What does the denoising function look like?
>
> **Answer to question1:**
> **Our depth control is integrated during the denoising stage during the DDIM Inversion process and the SDXL denoising process, in line 233-234 and 240-241 in the old version PDF.** We will clarify this point more clearly in our rebuttal response PDF.
>
> **The denoising function is:**
>
> $\epsilon_{t-1} = \epsilon_{\theta}(x_t, y, d)$
>
> Here, $\epsilon_{\theta}$ is the noise prediction model with depth controlnet, $x_t$ is the noise at step $t$, $y$ is the prompt and $d$ is the depth map, $\epsilon_{t-1}$ is the prediction.
>
> **Q2:** How to adjust the plane to be in contact with the object? How to deal with multiple planes?
>
> **Answer to question2:**
> 1. In the rendering process, a physical entity is required to capture shadows. Using the normal vector, we can automatically create a transparent plane in Blender to capture shadows without rendering the plane itself. This allows the shadows of the object to be accurately projected onto the plane, resulting in realistic lighting and shadow effects. **Even with relative depth, the normal vector's direction remains consistent**, enabling us to automate the calculation of the object's bottom coordinates for plane placement.
> 2. However, as noted in the limitations section of the supplementary materials, when the background plane is overly complex, our plane estimation method may not perform optimally. To address this, we have proposed potential solutions in the supplementary materials and future work. **Specifically, we suggest utilizing 3D reconstruction techniques to recreate the background environment, allowing for accurate rendering in more complex scenarios.**

---

> > ### Author Response · Authors · 2024-11-20
> >
> > **Q3:** How is the material property specified during rendering in the editing stage?
> >
> > **Answer to question3:**
> > **For material properties, most existing 3D generation methods primarily focus on mesh and texture mapping, without incorporating material attributes.** With advancements in 3D generation techniques, it is likely that more models will emerge with the capability to generate material-related attributes. We plan to closely follow these developments and explore how they can further enhance our framework.
> > As a result, our current approach focuses on optimization at the texture level.

---

> > > ### Comment · Reviewer_LJb3 · 2024-11-26
> > >
> > > Thanks for the rebuttal that addressed many of the questions.
> > >
> > > > In lines 301-303 of our old version PDF, we have described several methods for automating action binding. In the supplementary materials (line 776 of the old version PDF), we have provided a link to our anonymous Git repository, which demonstrates some human animation effects achieved through automated binding.
> > >
> > > It is non-trivial to do automatic rigging, skinning, and animation given an object mesh. Can the authors elaborate how it works, or mention which method was used to generate the results shown in the paper?
> > >
> > > > Answer to question1: Our depth control is integrated during the denoising stage during the DDIM Inversion process and the SDXL denoising process, in line 233-234 and 240-241 in the old version PDF. We will clarify this point more clearly in our rebuttal response PDF.
> > >
> > > It's still not clear how the depth conditioning works. It would be useful to elaborate or update in the revision. Is the SDXL fine-tuned with depth conditioning, or is depth added to the denoising process in a ad-hoc way without fine-tuning?

---

> ### Author Response · Authors · 2024-11-26
>
> **Q1**: It is non-trivial to do automatic rigging, skinning, and animation given an object mesh. Can the authors elaborate how it works, or mention which method was used to generate the results shown in the paper?
>
> **Answer to Q1:** Thank you very much for your question about automatic rigging, skinning, and animation. To make our work clearer, we will clarify these points as follows:
>
> 1. For human skeletons, we used platforms like [Mixamo](https://www.mixamo.com/#/) to upload meshes online, select skeletal key points, and let the system automatically rig the skeleton and generate corresponding animations based on the selected actions.
>
> 2. For animal skeletons, we utilized the built-in animal rigs and animations provided by Auto Rig Pro in Blender.
> 3. The results of our auto-animation process are showcased in Figure 9, where all animations were created using Mixamo. And for Figure 5, the pumpkin skeleton was manually rigged, while the elephant skeleton was rigged by Auto Rig Pro, and we performed a small amount of manual adjustment.
>
> We apologize for not being clear enough in our writing. We have provided a more detailed example in lines 335-337 to explain how automatic rigging works in our new revision PDF.
>
> **Q2:** It's still not clear how the depth conditioning works.
>
> **Answer to Q2:** We apologize if our explanation caused any misunderstanding. Our depth conditioning operates in two stages. The first stage involves DDIM inversion, and the second stage is during the denoising process without any fine-tuning. Depth control in both stages does not participate in training.
>
> Thank you very much for your comment. We specify the details of our depth control operations in lines 255-258 of the revised PDF.

---

> ### Author Response · Authors · 2024-12-02
>
> Thank you very much for your detailed responses and for acknowledging the contributions of our paper. We greatly appreciate your insightful feedback.
>
> We have carefully addressed your concerns and made revisions accordingly. Specifically, we have conducted the automatic rigging, skinning, and animation process in the revised paper. We’ve also clarified how the depth conditioning works.
>
> We hope that we have adequately addressed your concerns. If any issues remain unresolved, we are more than willing to further explain and clarify any points that may require additional attention. We look forward to your response.

---

> > ### Comment · Reviewer_LJb3 · 2024-12-03
> >
> > Thanks for the response, which addressed my comments.

---

> > > ### Author Response · Authors · 2024-12-04
> > >
> > > Thank you very much for your insightful comments. We highly appreciate your constructive feedback. Based on your suggestions, we have made the following revisions:
> > >
> > > We have added additional ablation studies in the supplementary materials to provide a clearer explanation of our approach.
> > > We have further clarified the explanation of the automatic rigging process in lines 335-337 of the revised PDF.
> > > The role of depth control has been made more explicit in lines 255-258 of the revised PDF.
> > > These improvements were directly influenced by your comments, and we believe they make our work clearer and more comprehensible.
> > >
> > > **If there are any remaining concerns, we would be more than happy to address them. We are also kindly wondering if there might be a possibility of raising the score once your questions have been fully addressed.**

---

### Official Review · Reviewer_oVkz · 2024-11-03

**Soundness:** 3
**Presentation:** 3
**Contribution:** 3
**Rating:** 5
**Confidence:** 4

**Summary:**

In this paper, the authors present a framework named OMG3D for object manipulation from images. Similar to Image Sculpting (Yenphraphai et al. 2024), OMG3D first segments the object and reconstruct a mesh in 3D. The mesh can then be mainuplated by 3D manipulation software. To address the loss of details in the rendered results, the authors propose a texture refinement module, named CustomRefiner, which performs gradient backpropagation directly to the UV texture map through differentiable rasterization. To achieve realistic lighting and shadow effects, the authors propose a lighting processing module, named IllumiCombiner, which estimates lighting from the background images and renders shadows that align with the scene.

**Strengths:**

+ Compared with Image Sculpting, OMG3D can produce results showing better texture alignment with the original image and achieve realistic light and shadow effects.
+ The idea of gradient backpropagation to the UV texture map through differentiable rasterization sounds novel.
+ The idea of estimating a spherical light from the background image and applying it in the rendering pipeline to achieve realistic shading and shadow sounds logical.
+ The qualitative results look convincing.

**Weaknesses:**

- A large part of the proposed pipeline comes from Image Sculpting. For instance, the "precise geometric control" is made possible by object segmentation followed by image-to-3D. The generative enhancement used in driving the UV-texture optimization also appears to be identical to that in Image Scuplting. This makes this work a bit incremental and lowers its novelty. Overall, this work can be regarded as an integration of Image Scultping (for 3D model manuipulation) and DiffusionLight (for introducing light and shadows effects).
- The key difference between this work and Image Sculpting is the replacing of direct rendered image enhancement with UV-texture optimization. However, the authors fail to discuss/analyze in detail why UV-texture optimization can produce better results than direct image enhancement.
- The light processing and plane creation part is not very clear.

**Questions:**

- In Fig.4, there are two "(ours)". It seems that the one in the 2nd column is a typo.
- How is c_a in (6) used to correct the color of the estimated lighting?
- What are I_d and I_c in (7)? How to adjust I_d "to maintain the object's saturation while ensuring that shadows remain evenly distributed in all direcionts"?
- How is the normal vector from the depth map being used in rendering shadows?
- Is the motion manually defined for the results of Image Scultping? How come the pumpkin jumping example shows lack of motion?

---

> ### Author Response · Authors · 2024-11-20
> ****Rebuttal to Reviewer oVkz****
>
> We deeply appreciate your recognition of the valuable insight behind our method and our presentations. We will address your inquiries and concerns point by point in the following responses.
>
> **W1:** How does the proposed method differ from Image Sculpting and DiffusionLight to ensure it is not incremental?
>
> **Answer to weakness 1:**
> We greatly appreciate the contributions of Image Sculpting(we call ImgScu in the following respose), as this work has been highly inspiring for our own approach.
>
> Our framework differs significantly from ImgScu and DiffusionLight in both motivation and methodology. Therefore, we respectfully disagree with the statement that our work can be regarded as an integration of Image Sculpting and DiffusionLight. Our reasons are as follows:
>
> 1. **Our motivation fundamentally differs from that of ImgScu.** The core motivation of our work lies in two aspects: first, achieving **realistic lighting and shadow effects** in both static and dynamic generation; second, ensuring consistent object appearance and smooth motion when applying 3D models directly **in dynamic generation**. In contrast, ImgScu primarily focuses on using 3D depth information to generate edited results and employing customized models to **refine 2D static outputs.** **Although both approaches incorporate certain 3D priors within their frameworks, it is inaccurate to classify our work as incremental to theirs.**
> 2. **We did not simply adopt existing lighting estimation techniques.** Our IllumiCombiner first estimates a spherical light, then uses the original image's color information to correct the estimated lighting colors, and finally adjusts light intensity to ensure the fidelity of the rendered results, as stated in lines 80–83 of the old version PDF. Furthermore, as shown in Figure 4 of the old version PDF, we conducted ablation experiments to demonstrate the effectiveness of our uniquely designed IllumiCombiner. **Simply using existing light estimation techniques would not achieve satisfactory results.**
>
> In summary, given our differing problem focus and methodological design, we respectfully disagree that our work is an incremental extension of ImgScu and DiffusionLight. We also acknowledge that our explanation here is not very clear. **In our new version of the PDF, we have added analysis in supplementary to address these points.**
>
> **W2:** Why does UV-texture optimization in OMG3D produce better results than direct image enhancement in ImgScu?
>
> **Answer to weakness2:**
> We will address the limitations of ImgScu in static editing and dynamic generation to **highlight the drawbacks of direct image enhancement.** Additionally, we will explain how UV-texture optimization can effectively avoid these issues and deliver improved results.
> 1. **First, we highlighted some limitations of ImgScu in static editing.** As discussed in our answer to Weakness 1, ImgScu relies solely on depth information during the generation process, which leads to issues such as **edge blurring and loss of details.** For instance, as shown in line 4 of Figure 4 in old version PDF, **the basketball in the hand of the right toy is missing.**
> 2. **Then, we highlighted the challenges of applying ImgScu to dynamic generation,** as noted in lines 40–42 of the old version PDF. While DreamBooth can improve the fidelity of 2D results, it cannot consistently or perfectly match the appearance of the original image. When applied to frame-by-frame editing, this inconsistency in object appearance can severely degrade video quality. Combined with the limitations in static editing, **this results in visual artifacts like flickering,** as shown in Figure 5 of the old version PDF.
> 3. **In contrast, our method directly renders the refined 3D object back to 2D.** Once the texture optimization is complete, the refined 3D model can be directly rendered for both static and dynamic scenarios **without requiring further refinement of the rendering results.** The physical rendering process in our method ensures object appearance consistency and motion continuity, effectively avoiding issues such as blurring and detail loss.  Moreover, the physical rendering process inherently supports the incorporation of light sources to achieve realistic lighting effects—**a capability ImgScu lacks. This fundamental difference underscores the significant advantage of our method over ImgScu.**
>
> Finally, we would like to once again express our gratitude to the authors of Image Sculpting for their valuable contributions to the community, which have greatly inspired our work. **And we have added these analyses into the supplementary materials in our revision PDF.**

---

> ### Author Response · Authors · 2024-11-20
>
> **W3, Q2, Q3, Q4:** How does the proposed method handle color correction, saturation adjustment, and shadow rendering using the estimated lighting and depth map?
>
> **Answer to weakness 3, question 2, question3, question4:**
> Thank you very much for your question about light processing and plane creation. Some concepts, such as the implementation of lighting effects and plane creation in Blender, as well as light color and intensity, are well-established within the field. These are explained in lines **264–289** of the old version PDF. **Your review has made us realize that these concepts may still be challenging for some to understand.** To make our work more accessible to a broader audience, we will clarify these points in our revisions as follows:
> 1. About plane creation:  In the rendering process, a physical entity is required to capture shadows. **With this normal vector, we can automatically create a transparent plane in Blender to capture the shadows without rendering the plane itself.**
> 2. About the lighting process: **We first correct the light color to ensure visual consistency between the rendered result and the original image,** as shown in Figure 4 and Section 4.3. Our estimated lighting $L_e$ has two properties: the color component $c_e$ and the intensity component $i_e$. We first use the original object's color $c_a$ to adjust the color of our estimated light $c_e$. This process can be defined as:
> $
> c_{\text{ec}} = \lambda_1 \cdot c_a + (1 - \lambda_1) \cdot c_e
> $
> Here, $c_{\text{ec}}$ is the corrected color.
>
> 3. We aim to maintain a base level of light brightness, preventing the rendered image from appearing too dark due to insufficient estimated intensity, and ensuring satisfactory color saturation. To achieve this, we define a uniform ambient light $L_d$ (with equal intensity $i_d$ in all directions and uniformly white in color), thereby inherently providing a base level of brightness. We then use $i_d$ to adjust $i_e$, the adjusted light intensity $i_{\text{ec}}$:
> $
> i_{\text{ec}} = \lambda_2 \cdot i_d + (1 - \lambda_2) \cdot i_e
> $
>
> We sincerely appreciate your question and recognize that there is room for improvement in our explanations. To address this, **we have clarified all these points in Sec 4.3 of our new version PDF.**

---

> > ### Author Response · Authors · 2024-11-20
> >
> > **Q1:** Is the "(ours)" label in the 2nd column of Fig. 4 a typo?
> >
> > **Answer to question 1:**
> > Thank you very much for pointing out the typographical error. We will correct the figure in our rebuttal response PDF.
> >
> > **Q5:** Is the motion manually defined for the results of ImgScu? How come the pumpkin jumping example shows lack of motion?
> >
> > **Answer to question 5:**
> > **We selected frames with more pronounced deformations for visualization.** The results of ImgScu don’t look like a continuous jumping motion but more like different pumpkins constantly deforming.
> > In the supplementary materials (line 776 of the old version PDF), we provide a link to our anonymous Git repository, **showcasing several dynamic results of ImgScu.**
> > **And we already adjust our frame selection in Figure 5 of the rebuttal response PDF** to more clearly highlight comparisons with our method.

---

> ### Comment · Reviewer_oVkz · 2024-11-25
>
> After reading other fellow reviewers' comments and also the authors' rebuttal, I would keep my initial rating.

---

> ### Author Response · Authors · 2024-11-27
>
> **We sincerely appreciate the time and effort you have taken to review our rebuttal.** We have worked diligently to address your concerns, thoroughly **analyzing the differences between our method and ImgScu.** Additionally, our experimental results demonstrate that our approach **indeed outperforms ImgScu.** We have also provided further clarification on the lighting process and plane creation and made corresponding revisions to the paper.
>
> We would like to know if there are any remaining questions or concerns that we have not yet addressed, or if there are specific experiments you would like us to conduct to further validate our conclusions. **We are committed to addressing all your concerns and ensuring our paper meets your expectations.**
>
> **If there are no further concerns, we would greatly appreciate it if you could consider updating your score, in line with the ICLR review process, once your questions have been fully addressed.** We hope that the additional clarifications and details we’ve provided have helped resolve any remaining doubts, **and we would be grateful if you would kindly reconsider your evaluation of our paper.**
>
> Finally, we are deeply grateful for your recognition of our work and sincerely hope that you can continue to see the value in our contributions. We truly believe that with your feedback, our paper will be greatly improved.

---

> ### Author Response · Authors · 2024-12-02
>
> Thank you very much for your responses and for acknowledging the contributions of our paper. We greatly appreciate your insightful feedback.
>
> We have carefully addressed your concerns and made revisions accordingly. Specifically, we have conducted analyses in detail between our method and ImgScu, with experimental results demonstrating that our approach outperforms ImgScu. We have also provided further clarification on the lighting process and the creation of planes, and have revised the paper to reflect these updates.
>
> We hope that we have adequately addressed your concerns. If any issues remain unresolved, we are more than willing to further explain and clarify any points that may require additional attention. We look forward to your response.

---

### Official Review · Reviewer_k8iy · 2024-11-03

**Soundness:** 3
**Presentation:** 3
**Contribution:** 3
**Rating:** 8
**Confidence:** 4

**Summary:**

This paper proposes a method for inserting 2D objects into 3D and letting users modify them. The method involves training a customized diffusion model added with a lighting processing module.

**Strengths:**

They can combine precise geometric control.
They can handle better texture renderings.
They can handle lighting better.

They offer complete comparison to showcase their results with other state-of-the-art methods.

**Weaknesses:**

They can try comparison with VSD loss. Or provide more visual examples in the supplementary results.

**Questions:**

Have you tried to insert real humans instead of animation? Want to see that result.

---

> ### Author Response · Authors · 2024-11-20
> ****Rebuttal to Reviewer k8iy****
>
> We deeply appreciate your recognition of the valuable insight behind our method and our presentations. We will address your inquiries and concerns point by point in the following responses.
>
> **W1 and Q1**: Show more comparison with VSD loss or provide more visual examples(Human animation).
>
> **Answer to weakness 1 and questions 1:**
>
> 1. Our comparisons primarily focus on image-to-3D generation methods; however, our framework can also incorporate text-to-3D approaches, such as [a] ProlificDreamer. **Additionally, we include comparisons that highlight cases utilizing VSD loss in Figure 7 and cite this work in our rebuttal response PDF.** Although text-to-3D methods like ProlificDreamer cannot directly model from images, we sincerely thank you for your inspiration.
> 2. In the supplementary materials (line 776 of the old version PDF), we have provided a link to our anonymous Git repository, which demonstrates several applications of our framework in human animation. Additionally, we add more visualization results in Figure 9 (the supplementary material of the new version PDF). **We apologize for not making it clear initially that the supplementary materials included linked content. To make our results clearer, we have now presented them in a more direct manner.** Furthermore, once we obtain a human model, we can leverage existing automated motion binding methods to bind human actions, enabling highly realistic and physically plausible human animations. This is also one of the advantages of our method.
> 3. **Thank you for your insightful feedback, which has been truly inspiring for us.** Your suggestions have motivated us to refine our explanations and make our paper more readable and clear. With your inspiration, we have included additional ablation studies on the various techniques used in our framework in Figure 8 of the supplementary material (new version of the PDF). These updates aim to provide clearer insights and a more comprehensive presentation of our method.
>
> Once again, we sincerely thank you for your positive recognition and support.
>
> [a] Wang Z, Lu C, Wang Y, et al. Prolificdreamer: High-fidelity and diverse text-to-3d generation with variational score distillation[J]. Advances in Neural Information Processing Systems, 2024, 36.

---

> ### Author Response · Authors · 2024-12-02
>
> Thank you very much for your detailed responses and for acknowledging the contributions of our paper. We greatly appreciate your insightful feedback.
>
> Regarding your concern about the comparison with existing method used the VSD Loss, as well as the demonstration of human animation, we have included these results in our revised PDF. Furthermore, inspired by your suggestions, we have added additional ablation studies to further enhance the clarity and comprehensiveness of our framework.
>
> We hope that we have adequately addressed your concerns. If any issues remain unresolved, we are more than willing to further explain and clarify any points that may require additional attention. We look forward to your response.

---

### Official Review · Reviewer_ee9t · 2024-11-04

**Soundness:** 2
**Presentation:** 3
**Contribution:** 1
**Rating:** 3
**Confidence:** 4

**Summary:**

The paper introduces OMG3D, a framework for object manipulation in images that addresses both static editing and dynamic motion generation challenges, while also improving the realism of object appearance and lighting. OMG3D integrates precise geometric control with the generative capabilities of diffusion models, converting 2D objects into 3D for user-directed modifications and lifelike motions. To enhance texture realism, it proposes CustomRefiner, a module that refines textures using a customized diffusion model to match the style and perspective of the original image. Additionally, the authors introduces IllumiCombiner, a module that adjusts background lighting to create more realistic illumination, aligned with human visual perception. The authors conducted extensive experiments show that OMG3D achieves good performance in both static and dynamic applications.

**Strengths:**

- The proposed method utilizes explicit 3D generation capability to ensure both static and dynamic manipulation. While other 2D-based methods fail to do so.
- The utilization of HDRi for realistic lighting.
- Visual quality is good.

**Weaknesses:**

- Lack of technical novelty, most of the presented techniques exist. The proposed method seems to put them together nicely to produce a few good results. E.g., CustomRefiner is a combination of depth-controlnet, dreambooth lora, and differentiable rendering on UV map, IllumiCombiner is a combination of HDRi estimation and virtual-plane rendering.
- The idea of realistic lighting is interesting to me, but I am not convinced by the proposed method. The real light transport is more complex than by linearly modulated two terms. Solving global illumination is a very difficult problem with a single image. The proposed method can only handle simple objects.

Minors:
- L216: IlluminCombiner should be bold.

**Questions:**

See above.

---

> ### Author Response · Authors · 2024-11-20
> **Rebuttal to Reviewer ee9t**
>
> We sincerely appreciate your positive remarks on our paper's visual results and the recognization of our realistic lighting methods.  Thank you very much for your valuable comments.
> Before addressing specific questions,  we want to highlight the significant contributions of our work, which have been well-recognized by other reviewers. Our key contributions include:
> 1. Our framework, OMG3D, **achieves both static edits and dynamic generation with precise geometric control**, delivering impressive visual results. This was recognized by Reviewer k8iy, Reviewer oVkz, and Reviewer LJb3.
> 2. **To ensure the fidelity of our results, we refine the coarse 3D generation results to achieve highly accurate texture matching with the original image.** This innovation has also been acknowledged and praised by Reviewer k8iy, Reviewer oVkz, and Reviewer LJb3.
> 3. We design the IllumiCombiner module to estimate and correct lighting, **enabling our framework to achieve realistic lighting and shadow effects.** We sincerely appreciate your acknowledgment of our idea to incorporate lighting into the generation process. This contribution was also recognized by Reviewer k8iy, Reviewer oVkz, and Reviewer LJb3.
> 4. Compared to other dynamic generation methods, **our approach is significantly more computationally efficient, requiring only a single NVIDIA 3090 GPU.** We are grateful that Reviewer LJb3 highlighted this practical advantage.
>
> **We will address your inquiries and concerns point by point in the following responses.**
>
> **W1:** Lack of technical novelty, most of the presented techniques exist.
>
> **Answer to weakness 1:**
> Thank you for your comment. We did not simply adopt existing techniques but instead introduced innovative designs tailored to the specific task. The novelty can be further summarized as follows:
> 1. Compared to existing texture optimization methods simply using reference image as guidance,  such as [a] DreamGaussian and [b] Repaint123, our CustomRefiner innovatively incorporates DreamBooth-LoRA, **which not only enabling customized generation that ensures consistent appearance with the original input but also significantly reduces computational cost.** Additionally, leveraging the advantages of 3D-based tasks, our framework provides precise depth information to assist in accurate geometric control.
> 2. While it is true that some existing techniques are incorporated within our framework, **the key contribution lies in how these techniques are adapted, redesigned, and integrated to address the specific challenges of our task.** Simply combining techniques like DreamBooth and Depth-ControlNet without our task-specific design would fail to achieve effective texture refinement or high-fidelity results.
> 3. **We did not simply adopt existing lighting estimation techniques.** Our IllumiCombiner first estimates a spherical light, then uses the original image's color information to correct the estimated lighting colors, and finally adjusts light intensity to ensure the fidelity of the rendered results, as stated in lines 80–83 of the old version PDF. **This approach is novel and has not been explored in previous or current generation frameworks.** Furthermore, as shown in Figure 4 of the new version PDF, our ablation experiments validate the effectiveness of this uniquely designed IllumiCombiner. **Simply relying on existing estimating techniques would fail to achieve satisfactory results.**
> 4. **To ensure readers clearly understand that our work is not merely a combination of existing techniques, we have added additional ablation studies about the techniques used,** as demonstrated in Figure 8 (the supplementary material of the new version PDF). These experiments provide comprehensive evidence of the necessity and efficacy of each technique within our framework. **Additionally, we have provided a more detailed explanation of our lighting processing innovations in Sec 4.3 of the new version PDF to enhance understanding.**
>
> In summary, our work does have significant novelty, but we sincerely appreciate your comment, and we will make our explanations clearer in the new version.
>
> [a] Tang J, Ren J, Zhou H, et al. Dreamgaussian: Generative gaussian splatting for efficient 3d content creation[J]. arXiv preprint arXiv:2309.16653, 2023.
>
> [b] Zhang J, Tang Z, Pang Y, et al. Repaint123: Fast and High-Quality One Image to 3D Generation with Progressive Controllable Repainting[C]//European Conference on Computer Vision. Springer, Cham, 2025: 303-320.

---

> ### Author Response · Authors · 2024-11-20
>
> **W2:** How does the proposed method address the complexity of real light transport and global illumination?
>
> **Answer to weakness 2:**
> 1. Thank you again for recognizing our idea of incorporating realistic lighting. **Our method offers a practical solution to this challenge and represents the most feasible approach currently available. To the best of our knowledge, no prior work has addressed adding realistic lighting to the 2D image editing process.** As noted in lines 45–48 of the old version PDF, existing generation methods rely solely on the model’s imagination for lighting effects. In contrast, our approach introduces lighting effects grounded in physical principles.
> 2. As shown in line 4 of Figure 3, previous methods struggle to achieve realistic lighting **even in simple scenes,** while our method succeeds in delivering convincing results.  **And we emphasize this point in line393 in our new version PDF.**
> 3. As shown in Figure 4 in new version PDF, our approach incorporates the state-of-the-art single-image environment light estimation method. Furthermore, as noted in lines 762–763 of the old version PDF, we acknowledge the need for further advancements in light estimation techniques. **We believe that as new and improved methods emerge, our framework is well-positioned to seamlessly integrate them, enabling the creation of even more complex and realistic lighting effects in the future.**
>
> Thank you again for recognizing our lighting method. We have made every effort to explore the possibilities to the best of our ability. In future work, we will actively investigate new developments to further enhance the performance of our framework.
>
> **Minor 1**: L216: IlluminCombiner should be bold.
>
> **Answer to minor 1:**
> Thank you for pointing out our typographical error, we make the correction in our rebuttal response PDF.

---

> > ### Comment · Reviewer_ee9t · 2024-11-21
> >
> > Thanks for the detailed responses.
> >
> > First, I agree that the contribution of this paper is **"how these techniques are adapted, redesigned, and integrated to address the specific challenges of our task"**, and I still have concerns about the technical novelty since the submitting venue is about **learning representation**. This paper may be more suitable for application-specific venues maybe?
> >
> > Second, **"To the best of our knowledge, no prior work has addressed adding realistic lighting to the 2D image editing process"**, while I oppose this claim. As far as I know, a lot of research work has been devoted to this area such as:
> >
> > * Photorealistic Object Insertion with Diffusion-Guided Inverse Rendering, ECCV 2024
> > * Diffusion Reflectance Map: Single-Image Stochastic Inverse Rendering of Illumination and Reflectance, CVPR 2024
> > * Intrinsic Harmonization for Illumination-Aware Compositing, SIGGRAPH Asia, 2023
> > * Neural Light Field Estimation for Street Scenes with Differentiable Virtual Object Insertion, ECCV 2022
> > * etc.

---

> > > ### Author Response · Authors · 2024-11-23
> > >
> > > **Q1:** Concerns about technical novelty.
> > >
> > > **Answer to Q1:** We sincerely appreciate your recognition of the contributions of our work, and we fully understand your concerns regarding technical novelty. Below, we would like to provide further clarification on this point:
> > >
> > > 1. **We believe that leveraging existing techniques to solve a new and challenging problem itself can be considered a form of technical novelty.** For instance, as demonstrated in [a] AnimateDiff (ICLR 2024 Spotlight), by combining existing Text-to-Image models with Temporal Attention, it successfully achieved high-quality video generation, garnering wide recognition from the community. This underscores that even when building on existing methods, careful design and integration tailored to specific challenges can drive substantial advancements in application domains.
> > >
> > > 2. **Our work's primary area is applications** to computer vision, audio, language, and other modalities. In addition to its application-oriented contributions, our method also **integrates core principles of representation learning,** **leveraging learned features from 2D tasks to enhance the quality and efficiency of 3D generation**, providing a complete and feasible solution. This not only fills existing research gaps but also lays a solid foundation for future exploration in this application domain.
> > >
> > > Thank you very much for your suggestions. **To clarify our technical novelty more effectively, we revised the description of our key contribution in lines 92–95 of the new revision PDF.** We specifically emphasized how our framework addresses the new challenges of static and dynamic generation by integrating a combination of advanced techniques.

---

> ### Author Response · Authors · 2024-11-23
>
> Q2: Overclaim. "To the best of our knowledge, no prior work has addressed adding realistic lighting to the 2D image editing process"  A lot of research work has been devoted to this area.
>
> **Answer to Q2:** We sincerely apologize if our statement seemed like an overclaim. Thank you for pointing this out and providing additional references to clarify this aspect. **What we intended to express is that existing methods utilizing diffusion denoising processes for image or video generation (without 3D prior) have lacked explicit control over lighting.** And our opinions are as follows:
>
> 1. It is really difficult for us to cover all relevant works, and we are deeply grateful for you bringing these works to our attention. We have carefully reviewed them and identified differences compared to our approach:
>
> [b] Wang's work first estimated an HDR sky map and then used Differentiable Object Insertion to integrate virtual objects with the background, though their work **focuses on outdoor scenes.**
>
> [c] Chris Careaga's work adjusted the foreground albedo to match the background and then estimated the environmental light, refining the shading process. But the albedo information of the objects is difficult to obtain, using a network to estimate albedo can introduce errors. As mentioned in their limitations, **their method hardly model the cast shadows that the object may generate in the new environment.**
>
> [d] Yuto Enyo proposed a method that leverages 3D object information to generate high-quality Reflectance Maps, thereby recovering detailed lighting information. Their shown results are more focused on applications to **objects with high reflectivity.**
>
> [e] DiPIR effectively estimate environmental light from background, which **requires per-object optimization for environmental lighting.** And our framework aims to enable seamless **combinations of arbitrary foregrounds and backgrounds.** Specifically, our IllumiCombiner leverages the recent work [f]DiffusionLight (CVPR 2024), which provides accurate environmental lighting estimation based on the input image, facilitating broader applicability of our approach.
>
> 2. **The first and most critical contribution of our work, OMG3D, lies in proposing a framework that enables seamless integration of static edits and dynamic generation through 3D object manipulation.** To accomplish this primary goal, we introduce a series of subtasks, such as texture optimization to enhance reconstruction fidelity and environment lighting estimation and correction to achieve realistic lighting effects. **In this process, each module within our framework is designed to be plug-and-play.** **The methods proposed in the papers you mentioned can serve as alternative implementations** for the lighting estimation module within our framework, but they do not diminish the originality or overall contribution of our work. Instead, **they highlight the flexibility and adaptability** of our framework to incorporate state-of-the-art methods.
>
> **We have added our analysis of these lighting methods in lines 144-161 of our revised PDF and have cited these works accordingly.** [e] DiTIR represents a state-of-the-art approach for optimizing environment lighting. **We will incorporate this work into our framework as soon as their implementations are made publicly available.**
>
> [a] Guo Y, Yang C, Rao A, et al. Animatediff: Animate your personalized text-to-image diffusion models without specific tuning[J]. arXiv preprint arXiv:2307.04725, 2023.
>
> [b] Wang Z, Chen W, Acuna D, et al. Neural light field estimation for street scenes with differentiable virtual object insertion[C]//European Conference on Computer Vision. Cham: Springer Nature Switzerland, 2022: 380-397.
>
> [c] Careaga C, Miangoleh S M H, Aksoy Y. Intrinsic Harmonization for Illumination-Aware Compositing[J]. arXiv preprint arXiv:2312.03698, 2023.
>
> [d] Enyo Y, Nishino K. Diffusion Reflectance Map: Single-Image Stochastic Inverse Rendering of Illumination and Reflectance[C]//Proceedings of the IEEE/CVF Conference on Computer Vision and Pattern Recognition. 2024: 11873-11883.
>
> [e] Liang R, Gojcic Z, Nimier-David M, et al. Photorealistic object insertion with diffusion-guided inverse rendering[J]. arXiv preprint arXiv:2408.09702, 2024.
>
> [f] Phongthawee P, Chinchuthakun W, Sinsunthithet N, et al. Diffusionlight: Light probes for free by painting a chrome ball[C]//Proceedings of the IEEE/CVF Conference on Computer Vision and Pattern Recognition. 2024: 98-108.

---

> > ### Comment · Reviewer_ee9t · 2024-11-26
> >
> > Thank you for the detailed responses and for taking the time to address my concerns. While I appreciate the clarifications and the additional information provided, my fundamental concerns regarding the technical novelty, positioning of the paper, and the breadth of related work remain unresolved. Below, I outline my key points:
> >
> > - I recognize that the authors have made an effort to adapt and integrate existing techniques for the specific challenges outlined in the paper. However, the core contributions still lack sufficient novelty for a venue focused on representation learning. As the authors themselves acknowledge, much of the work is based on combining existing methods (e.g., DreamBooth-LoRA, Depth-ControlNet, HDRi estimation, etc.). While the integration is tailored, it does not substantially advance the state-of-the-art in learning representations or propose fundamentally new methods.
> > - The references I provided clearly demonstrate that significant prior work exists in this area, addressing challenges such as photorealistic object insertion, inverse rendering, and lighting estimation. While the authors argue that their IllumiCombiner module is novel, the overall contribution in this area feels incremental.
> >
> > In summary, while the paper achieves reasonable visual quality and integrates several interesting techniques, it does not present sufficient technical novelty or foundational contributions for this venue. The overclaim regarding realistic lighting and the incomplete literature review further detract from the overall rigor of the work. I don't see strong evidence to overturn my evaluation, thus I maintain my initial rating.

---

> ### Author Response · Authors · 2024-11-27
>
> We sincerely appreciate the time and effort you have taken to review our rebuttal. However, we respectfully disagree with your points, and we would like to present our perspective as follows:
> 1. We believe that our work presents significant technical novelty and **is well-suited for ICLR.**  Rather than simply utilizing existing methods, we have developed a new 3D learning representation, addressing the challenge of achieving realistic 3D reconstruction and rendering. This is achieved by combining 2D customized models that capture the visual features of the original object to generate new 3D views. We then optimize 3D textures with these presentations to achieve high-quality visual results.
>
>     Moreover, as we pointed out in our previous response, **ICLR 2024 has already recognized several works that integrate existing technologies to complete new tasks.** For example, **AnimateDiff (ICLR 2024 Spotlight)** combines a text-to-image diffusion model with temporal attention to generate high-quality video, and **3D-DST (ICLR 2024 Spotlight)** generates edge maps from virtual 3D object renderings as control and uses LLMs to expand prompts for guiding 2D generation. **These works, like ours, demonstrate how combining established methods can lead to meaningful advancements in the field.** In this context, we believe that our work, which leverages existing techniques to create a new, customized 3D learning representation, **fits well within the scope of ICLR** and contributes novel insights to the research community.
>
> 2. Our core innovation is centered around the framework itself, with the technical aspects providing additional support. **The key novelty of our work lies in a framework using geometric control to enable image editing and video generation.** This is the main contribution of our work. **During the development of this framework, we faced challenges** such as low fidelity in 3D reconstruction and a lack of lighting and shadow effects in the rendered results. In response, we introduced specific technical innovations to address these issues.
>     As a result, our framework is able to achieve excellent visual effects, which you have acknowledged. Additionally, compared to existing video generation methods, our framework is more cost-efficient—it only requires a single NVIDIA 3090 for execution. **We firmly believe that our framework offers significant innovation, rather than being an incremental improvement.**
>
> 3. We sincerely appreciate your suggestions and the additional lighting methods you recommended. We have indeed referenced and analyzed these techniques in our work.  However, in comparison to existing lighting methods, our IllumiCombiner provides a substantial improvement over more recent methods, such as **DiffusionLight (CVPR 2024)**, as demonstrated in our ablation study on lighting(Figure 4). We believe that **our approach to lighting design strikes a balance between effectiveness and efficiency**, tailored specifically to the objectives of our framework.
>
>     **That being said, even if our work were considered incremental, the lighting component should not be seen as the core contribution of our paper. Therefore, it would be unfair to dismiss the overall value of our work simply because our lighting method has not yet reached the current state-of-the-art level.**
>
> Thank you again for your thoughtful comments. We recognize that there is room for improvement in our writing, particularly in the lighting aspect.  We have revised the paper to clarify these points and strengthen our discussion. **Additionally, we plan to include a more detailed analysis of the lighting in the limitations section of the supplementary material, where we’ll explore potential improvements in future work.**

---

> ### Author Response · Authors · 2024-12-02
>
> Thank you very much for your detailed responses and for acknowledging the contribution of our paper. We greatly appreciate your insightful feedback.
>
> Regarding your concern about the technical novelty of our work, we would like to emphasize once again that our core contribution is indeed technically novel. We have also provided examples from ICLR Spotlight 2024, such as AnimateDiff and 3D-DST, to demonstrate that these methods can lead to meaningful advancements in the field, and that our work is well-suited for ICLR. We sincerely appreciate your suggestions and the additional lighting methods you recommended. We have indeed referenced and analyzed these techniques in our work.
>
> We hope that we have adequately addressed your concerns. If any issues remain unresolved, we are more than willing to further explain and clarify any points that may require additional attention. We look forward to your response and are eager to continue the discussion.

---

### Author Response · Authors · 2024-11-25
**Looking Forward to Your Feedback**

Dear Reviewers,

We are highly grateful for your time and effort to review our work! We understand that you may have busy schedules, but we greatly value your feedback. And we are pleased that reviewers are excited about the novel contributions of our work. Reviewers remark that **“the idea of realistic lighting is interesting to me”**[ee9t],  **“they can combine precise geometric control, handle better texture renderings and handle lighting better.”**[k8iy],  **“the idea of UV texture map optimization sounds novel”**[oVkz], **“the paper is well written and the qualitative results are well made”**[LJb3]. We thank the reviewers for the strong praise of our work and contributions.

We now highlight a few important points raised by reviewers in the general response:

### Highlights of New Analyses and Experiments

In response to reviewers' comments, we have included these analyses and experiments in individual comments and revision PDF.  We now briefly describe these responses.

* **Explanation of core technical novelty and analyses of lighting methods** [ee9t]:  In our introduction of revision PDF, we have emphasized that the core contribution of our framework is the proposal of an application framework. Additionally, we have updated the related work section to include comparisons and analyses of existing lighting estimation methods.

* **More visual results and comparisons** [k8iy]: In the supplementary materials, we have added more visual results of human animation and provided additional comparisons with text-to-3D generation methods using VSD loss.

* **More analyses with Image Sculpting**[oVkz]: We have added more analyses in the comments and supplementary materials to highlight the differences in motivation and methodology between our approach and the Image Sculpting method, emphasizing the distinctions of our work and why our OMG3D works better.

* **Clearer descriptions of the lighting process and plane creation** [oVkz]: We have revised the descriptions of the lighting process and plane creation in the updated PDF to make them clearer and more precise.

* **More ablation study of techniques we used**[LJb3]: We have included additional ablation studies in the supplementary materials to better demonstrate the importance of the techniques used in our work.

### Enhanced Figure

We updated Figure 4 and Figure 5 to improve clarity. We updated Figure 7, Figure 8 and Figure 9 to show more ablation studies and visual results.

We thank all reviewers for their thoughtful commentary. We worked hard to improve our paper, and we sincerely hope the reviewers find our responses informative and helpful. If you feel the responses have not addressed your concerns to motivate increasing your score, we would love to hear what points of concern remain and how we can improve our work. Thank you again!


Best regards,

The Authors

---

### Meta-Review · Area_Chair_o4o9 · 2024-12-14

**Metareview:**

This paper introduces a framework to manipulate single objects in existing images by geometry changes or animations. The method separates the image into foreground and background, before processing each part individually. From the background, the lighting is estimated to obtain an editable representation. The object is transformed into an explicit 3D representation that can be manually modified. In the end, the individual parts are composed into an image by a conditional diffusion model.

The main strengths of this work are the well-designed pipeline and good qualitative results. It seems that the method is able to capture changing lighting and geometry well and produces better results than the used baselines.

The main concern of me and also three out of four reviewers is the missing technical novelty. The individual parts of this work are taken and adapted from previous work. The main contribution of this work is mostly the overall applied system.

The reviews ended up with diverging borderline scores of 3/5/6/8, where the negatively leaning reviewers provided the stronger arguments in discussions. Since I agree with the arguments of lacking technical novelty, I tend to follow with a reject recommendation for this work. In case of rejection I strongly encourage the authors to submit this work to a more application focused venue, where it might be more positively received due to the high quality results.

**Additional Comments On Reviewer Discussion:**

After the first round of reviews, the paper already had borderline scores. The authors tried to address the concerns of negatively leaning reviewers but they were not convinced to increase their scores. The main concern was lacking technical novelty, which is hard to rebut.

At the same time, none of the positively leaning reviewers stood out to champion this paper, with the most positive one providing a lower quality review and no contributions to discussions.

---

### Decision · Program_Chairs · 2025-01-22

Reject